# Investigations on the Efficacy of Ozone as an Environmental Sanitizer in Large Supermarkets

**DOI:** 10.3390/pathogens11050608

**Published:** 2022-05-23

**Authors:** Giuseppina Caggiano, Marco Lopuzzo, Valentina Spagnuolo, Giusy Diella, Francesco Triggiano, Marilena D’Ambrosio, Paolo Trerotoli, Vincenzo Marcotrigiano, Giovanna Barbuti, Giovanni Trifone Sorrenti, Pantaleo Magarelli, Domenico Pio Sorrenti, Christian Napoli, Maria Teresa Montagna

**Affiliations:** 1Interdisciplinary Department of Medicine, Hygiene Section, University of Bari Aldo Moro, Piazza G. Cesare 11, 70124 Bari, Italy; giusy.diella@uniba.it (G.D.); paolo.trerotoli@uniba.it (P.T.); mariateresa.montagna@uniba.it (M.T.M.); 2Department of Biomedical Science and Human Oncology, University of Bari Aldo Moro, Piazza G. Cesare 11, 70124 Bari, Italy; marco.lopuzzo@uniba.it (M.L.); valentina.spagnuolo@uniba.it (V.S.); francesco.triggiano@uniba.it (F.T.); marilena.dambrosio@uniba.it (M.D.); giovanna.barbuti@uniba.it (G.B.); 3Department of Prevention, Food Hygiene and Nutrition Service, Local Health Unit BT, Barletta-Andria-Trani, 76125 Trani, Italy; vincenzo.marcotrigiano@aslbat.it (V.M.); giovanni.sorrenti@aslbat.it (G.T.S.); pataleo.magarelli@aslbat.it (P.M.); domenico.sorrenti@outlook.it (D.P.S.); 4Department of Medical Surgical Sciences and Translational Medicine, Sapienza University of Rome, 00189 Rome, Italy; christian.napoli@uniroma1.it

**Keywords:** ozone, environmental sanitizer, disinfection, fungi, bacteria, air, surface

## Abstract

Awareness of the importance of the microbial contamination of air and surfaces has increased significantly during the COVID-19 pandemic. The aim of this study was to evaluate the presence of bacteria and fungi in the air and on surfaces within some critical areas of large supermarkets with and without an ozonation system. Surveys were conducted in four supermarkets belonging to the same commercial chain of an Apulian city in June 2021, of which two (A and B) were equipped with an ozonation system, and two (C and D) did not have any air-diffused remediation treatment. There was a statistically significant difference in the total bacterial count (TBC) and total fungal count (TFC) in the air between A/B and C/D supermarkets (*p* = 0.0042 and *p* = 0.0002, respectively). Regarding surfaces, a statistically significant difference in TBC emerged between A/B and C/D supermarkets (*p* = 0.0101). To the best of our knowledge, this is the first study evaluating the effect of ozone on commercial structures in Italy. Future investigations, supported by a multidisciplinary approach, will make it possible to deepen the knowledge on this method of sanitation, in light of any other epidemic/pandemic waves.

## 1. Introduction

Interest in assessing the microbial contamination of air and surfaces increased significantly during the COVID-19 pandemic, revealing the importance of environmental microbiological monitoring systems as fundamental elements in response to epidemics, and applying specific methods of environmental control in addition to routine microbiological checks [1]. There has been a greater awareness of how microorganisms (bacteria, fungi and viruses) present in the air and on surfaces can have harmful effects on the health of individuals, and can spread very easily in public spaces, especially if they are closed and crowded, such as supermarkets. Therefore, good sanitation practices of surfaces and indoor air are actually considered as a prerequisite for the control and prevention of airborne infectious diseases.

Different approaches have been developed to reduce viable microbial counts in the air, including chemical aerosolization, ozonation and ultraviolet (UV) irradiation [2]. Although physical methods such as UV or pulse light are advantageous for inactivating microorganisms in indoor environments due to the lack of residues, ozone generators have recently assumed an important role due to their efficient penetration in inaccessible places [3]. Some authors indicate ozonation as an alternative method to inactivate airborne microorganisms, especially molds in cheese ripening and storage plants [4], foods that are easily exposed to fungal contamination [5]. Other authors [6] report the synergistic effect of some air treatment systems associated with ozonation.

In May 2020, the National Institute of Health [7] published some recommendations on the sanitation of non-health environments during the COVID-19 pandemic. Among the various disinfectants, ozone is mentioned by International Organizations, such as the European Chemistry Agency (ECHA), Centers for Disease Control and Prevention (CDC), Food and Drug Administration (FDA), United States Environmental Protection Agency (US-EPA). In addition, the International Ozone Association [8] confirms the effectiveness of ozone for the inactivation of many microorganisms, including viruses.

Ozone, an allotropic form of oxygen, is an inorganic gas made up of three oxygen atoms (O_3_), which easily decompose into oxygen (O_2_), and a single highly reactive oxygen atom. Ozone occurs in nature with an atmospheric concentration of approximately 0.04 parts per million (ppm: 1 ppm~2 mg/m^3^) and is also produced by the ultraviolet irradiation of oxygen or other precursors, such as the volatile organic compounds and nitrogen oxides present in the atmosphere. Ozone is known for its protective role in Earth’s ecological environment and it is a powerful oxidant, which is able to react with organic molecules containing double or triple bonds. This explains its bactericidal, virucidal and fungicidal properties, exploited in water treatment and medicinal applications. Ozone’s effectiveness on pathogenic microorganisms has been known since the 19th century [9,10].

Currently, in Italy, ozone can be used exclusively as a sanitizer. Ozone is presently being reviewed by the European Environmental Agency and/or Switzerland for use as a biocide, in disinfection, food and animal’s feeds, drinking water and as a preservative for liquid systems, under the Biocidal Products Regulation of ECHA [7,9]

The aim of this study was to evaluate the presence of bacteria, molds and yeasts in the air and on surfaces within some critical areas of large supermarkets with and without an ozonation system. 

## 2. Results

The median of TBC in the air of A/B supermarkets was 28 cfu/m^3^ (range 0–83) at 8:00 and 72.5 (range 44–170) at 12:00; the median of TBC in C/D supermarkets was 93 (range 39–264) at 8:00 and 87 (range 44–183) at 12:00. There was a statistically significant difference in TBC (Table 1) between A/B and C/D supermarkets (*p* = 0.0042), but there were no statistically significant differences for sale areas (*p* = 0.3924). The difference between times of sampling was statistically significant in supermarkets (*p* = 0.0052); the post hoc analysis showed that at 8:00 am, TBC was significantly higher in C/D supermarkets, but there was no statistically significant difference at 12:00 pm. 

The median of TFC in the air of A/B supermarkets was 25 (range 0–440) at 8:00 and 17 (range 0–50) at 12:00; in C/D supermarkets, it was 46.5 (range 11–180) at 8:00 and 85.5 (range 28–344) at 12:00. The difference in TFC between A/B and C/D supermarkets was statistically significant (*p* = 0.0002), but not for sale areas (*p* = 0.45). The difference between times of sampling was statistically significant in supermarkets (*p* = 0.0078), and the post hoc comparison showed a statistically significant difference, with a higher TFC in C/D supermarkets at 12:00 but not at 8:00. As regards the isolated species, the filamentous fungi predominated. In particular, in A/B supermarkets in all sampling, *Penicillium* was the most frequently isolated species (93%; 28/30 positive samples), followed by *Feifomycetes* (40%; 12/30) and *Aspergillus* (16%; 5/30). Additionally, in ozone-free supermarkets (C/D), *Penicillium* was the most frequently isolated species (65%; 26/40 positive samples) followed by *Mucorales* (25%; 10/40), *Feifomycetes* (22.5%; 9/40) and *Aspergillus* (15%; 6/40).

Regarding surfaces, the median of TBC in A/B supermarkets was 0 cfu/m^3^ (range 0–4) at 8:00 and 0 cfu/m^3^ (range 0–2) at 12:00; the median of TBC in C/D supermarkets was 8.5 cfu/m^3^ (range 0–114) at 8:00 and 9.5 cfu/m^3^ (range 0–185) at 12:00. There was a statistically significant difference in TBC on surfaces (Table 2) between A/B and C/D supermarkets (*p* = 0.0101), but there were no statistically significant differences for sale areas (*p* = 0.1673). The difference between times of sampling was not statistically significant (*p* = 1).

On surfaces, the median of TFC in A/B supermarkets was 0.5 cfu/swab (range 0–12) at 8:00 and 0 cfu/swab (range 0–5) at 12:00; in C/D supermarkets, it was 1.5 cfu/swab (range 0–14) at 8:00 and 1 cfu/swab (range 0–5) at 12:00. The difference in TFC and sale areas between A/B and C/D supermarkets was not statistically significant (*p* = 0.2865 and *p* = 0.2556, respectively). The difference between times of sampling was not statistically significant (*p* = 1). Relating to the isolated species, on surfaces, only *Penicillium* was isolated in all 34 the positive samples for fungi.

## 3. Discussion

In the past, ozone was tested for the preservation of different food and/or food ingredients, such as milk and meat products [11], and for the purification and artificial aging of alcoholic beverages [12]. However, most known applications dealt with municipal and industrial wastewater [13] and the treatment of drinking water [14]. Interest in the use of ozone in the food industry was accompanied by the US government’s approval of ozone for safe use, in both the gaseous and aqueous phases, as an antimicrobial agent for foods [15]. More recently, some authors reported that ozone should be effective in treating contaminated surfaces and for the survival fraction of microorganisms through an exponential relationship, the results of which are influenced by the species of microorganisms, relative humidity present in the environment and used ozone dosage [16]. Other authors showed that, when evaluating the internal aspects of workplaces where food is regularly sold, it is worth considering the impact of ozone interactions on indoor air chemistry [17]. In fact, through a modeling study, the authors showed that approximately 80% of ozone is lost by deposition on surfaces. Today, the use of ozone attracts widespread attention from researchers to improve the shelf-life and safety of food products [18]. Similarly, available data relating to aquatic environments suggest that ozonation is able to increase the inactivation effect of microorganisms by approximately 50% in the log range [19]. 

Through touch-screen displays connected to the air conditioning and mechanical ventilation system control units, microclimatic characteristics were set and detected inside the four points of sale. In particular, a temperature between 20 and 22 °C was set and detected inside the four points of sale. However, these systems were not directly able to modify the quantity of relative humidity in the indoor environment, although was between 45 and 50%. Further, although the light intensity was not detected at the time of environmental sampling, supermarkets with ozone had cold light illumination fingers while ozone-free supermarkets were illuminated with warm light.

Our study examined TBC and TFC present at different times of the same day in supermarkets with and without ozone treatment (A–B vs. C–D). The data obtained from the air highlighted the presence of statistically significant differences in both TBC and TFC between the A–B and C–D supermarkets concerning the sampling times, with the exception of the sales areas. The difference in TFC in the air between 8 and 12 h was statistically significant and is probably related to a lower TFC at 12 in supermarkets with ozone sanitizers (A–B). It should be emphasized that the fungal charge in A–B supermarkets was similar at different times, while the C–D supermarkets showed significant variations between the sampling times. This could depend on many factors and not only on the absence of sanitation. In fact, as detected by the interquartile intervals, it can be hypothesized that the charge can be very low or very high in relation to the point of the sample (random variability) rather than to the type of structure. These results highlight the need to deepen the research on fungi both by increasing the sampling points and by increasing the facilities.

The dynamic change in microbial load in supermarkets or other commercial establishments has been reported [20,21], but to our knowledge, this is the first study that compare both bacterial and fungal contamination in supermarkets equipped with ozonation systems and supermarkets without.

In particular, Lee et al. [20] reported the results after the installation of fixed in-room ultraviolet germicidal irradiation air cleaners in commercial buildings, including restaurants and offices, demonstrating the significant reductions in both airborne and surface-borne bacterial contamination, even if the same authors pointed out that occupancy levels of commercial establishments varied according to their location, representing a limitation of the study.

Differently, Boonrattanakij et al. [21] investigated air and surface quality improvement in terms of both bacterial and fungal contamination in an exercise room of a fitness center. They monitored the microbiocidal disinfection efficiency on bioaerosols present in the fitness center under normal operating conditions, applying ClO_2_ and weak acid hypochlorous water, to air purification, and also ZnO, or commercial disinfectant to clean all contact surfaces of each equipment after each of the exercise session, resulting ClO_2_ the most appropriate and effective disinfectant. However, the study was carried out during the time of COVID-19 pandemic; thus, the number of users drastically was reduced.

Surface surveys also showed a statistically significant difference in TBC between supermarkets A/B and C/D, but no statistically significant difference emerged between the time of sampling and the sales areas. As regards the findings relating to TFC on surfaces, the differences between different supermarkets were not statistically significant, either for sampling time or for sales areas. 

Mold spores are spread ubiquitously in the environment, they are main vehicles for distribution of filamentous fungi in space via water, air [22]. The distribution of mold is almost unavoidable and in enclosed space, there are several limitations to the control of mold growth with respect to cleaning, ventilation. In our study genera *Penicillium, Aspergillus* were predominant. They are probably related to various factors, such as the movement of people inside the supermarket, opening and closing of doors, storage of products on the shelves as well as the presence of fruit and vegetables. However, it is important to underline that some fungal strains are developing resistance to the most widely used fungicides. 

The contamination of the surfaces was relatively low compared to that from the air. Since, the different surfaces, support tops and areas most frequently touched by customers were cleaned and sanitized during the day by dedicated staff, so the microbial contamination of the surfaces was probably less affected by the presence or absence of the ozonation procedure. So, it is possible that this makes the effect of ozone treatment on the air much more evident than on the surfaces, both as regards TBC and TCF. During the particular epidemiological situation, such as the current SARS-CoV-2 pandemic era and the well-known spread of other pathogens (e.g., Gram-negative and Gram-positive bacteria, molds, and yeasts) by air and/or by contact [23,24,25,26], special attention on the part of Public Health is essential. The microorganisms can lead to an increased risk of infection for the population, while adopting appropriate behaviors (e.g., physical distancing, hand hygiene or the use of personal protective equipment). Consequently, environmental treatment through the use of ozone is one of the remediation measures that has been evaluated in the COVID-19 era [27]. Compared to other disinfectants, lower ozone concentrations and shorter contact times are sufficient to inactivate the environmental microbial population [15]. Some authors state that ozone is also effective against viruses present on surfaces and in the air [28]. However, the use of ozone as a disinfectant has some disadvantages, e.g., it is extremely unstable and quickly reverts to a normal oxygen molecule (O_2_); therefore, it cannot be easily stored or transported [15]. 

Although our study showed good results in indoor food environments, we are aware that some limitations characterize our research. Few supermarkets were examined, although they are among those most frequented by the resident population; furthermore, the transversal design of the study limits its validity in the evaluation of the temporal relationship between exposure and outcome. 

Furthermore, in this preliminary study, two sampling times were considered, at the opening and during the most representative time slot of daily activity. The closing time was not taken into consideration as the treatment system was in operation from 8 am until closing. However, further studies could be carried out, evaluating the change during different working hours, different days (midweek and holidays), different seasons, although the indoor microclimatic conditions in food businesses are rather conserved during the calendar year.

During the numerous samplings carried out at the four points of sale investigated, it was not possible to determine the concentration of airborne ozone in a punctiform manner by means of portable devices for measuring the O_3_ concentration. Therefore, this preliminary study offers a further point of investigation, as it would be interesting to evaluate the concentration during the whole day both in the air and on surfaces. Moreover, it could also be interesting to consider whether the light intensity of the sampling sites of supermarkets affects the efficacy of the ozone, possibly affecting its degradation. Despite these potential biases, to the best of our knowledge, this is the first study evaluating the effect of ozone on commercial structures in Italy. It is our intention to complete this study with the search for SARS-CoV-2 in the same supermarkets and under the same conditions, having already found the virus in other community environments [29,30].

## 4. Materials and Methods

### 4.1. Study Design

Surveys were conducted in four supermarkets belonging to the same commercial chain, selected from the most representative ones of an Apulian city in June 2021. Of these, two (hereinafter referred to as A and B) were equipped with an ozonation system, and two (hereinafter referred to as C and D) did not have any air-diffused remediation treatment.

The ozonation systems in the two supermarkets are equipped with two or three generators (GX model, Ozotek^®^, Taranto, Italy), with closed corona discharge reactors, located in false ceilings with branches for the distribution of ozone in the selling areas. By means of microcompressors, the machines produce approximately 2.64 g ozone per hour. During the night, in the absence of staff or customers, cycles of ozonation are carried out for a total of 2 hours and 30 minutes. During the opening hours, microcycles of 10 minutes are performed every 2 hours, from 8:00 am to 6:00 pm. The staff can access workplaces at 7:00 am, and shops are opened to customers at 8:00 am. 

The sampling was carried out over four consecutive midweek days (one for each supermarket) not holidays, in order to reproduce conditions as reproducible as possible and with medium turnout. The areas most frequented by customers were selected for air sampling: bread, cheeses, cured meats, fish, fruit, vegetables and checkout areas; for surfaces sampling, scales for self-service products, refrigerator handles for beverages and dairy products, freezer handles for ice cream, cash register keyboards used by cashiers and POS keyboards used by customers were selected.

The samples were taken at opening hours (8:00 am) and at the most representative time of the high levels of customer use (12:00 pm), transported to the laboratory in a suitable isothermal refrigerator at a controlled temperature (4 °C) and immediately processed. Overall, 10 air and 20 surface samples were examined for each supermarket both at 8:00 am and at 12:00 pm, for a total of 240 samples (80 of air and 160 of surfaces).

### 4.2. Environmental Sampling

For each supermarket, indoor air was sampled using the SAS (Surface Air System, SAS Super ISO 180; PBI International, Milan, Italy). We used an active sampling method on solid substrates that conveys the aspirated air (180 L/min) directly onto 55 mm Petri dishes containing Plate Count Agar (PCA; Becton–Dickinson, Heidelberg, Germany) for bacterial count and Sabouraud dextrose agar with 0.05% chloramphenicol (BioRad, Marnes-la-Coquette, France) for mycological (molds and yeasts) investigations. Filamentous fungi were identified by evaluating standard cultural characteristics (e.g., morphology, colony color, surface growth type and macro- and microscopic examination) [24,25,26], and with a biochemical technique through the FF MicroPlate of Biolog (Rigel Process and Lab, Motta Visconti Milan, Italy). This consisted of employing redox chemistry based on the reduction in tetrazolium, responding to the process of metabolism (oxidation of substrates).

The surface sampling was performed using sterile swabs moistened with 2 mL of sterile physiological solution. On flat surfaces, the swab was rotated on a standard-sized sampling area (10 cm × 10 cm); on non-flat surfaces (handles and keyboards), the swab was rotated on the most exposed zone. 

After incubation at 30 ± 1 °C for 72 h for bacteria research and at 28 ± 1 °C for 8 d for mycological valuation, the air results were expressed as the number of colony forming units/m^3^ (cfu/m^3^) and the swab results as cfu/swab.

### 4.3. Statistical Analysis 

The bacterial and fungal counts were synthetized as the median and range by hours and supermarket type (with or without ozone). To compare the total bacterial count (TBC) and total fungal count (TFC), which were not normally distributed, they were transformed into ranks and then analyzed in a general linear model to compare supermarkets with and without ozone treatment, and hours of sampling. The model used bacteria or mycological counts as the dependent variables and the type of supermarkets (with or without ozone), sale areas, hour of sampling (T1 = 8:00; T2 = 12:00) and interaction type/hour as the independent variables. Post hoc comparisons were performed using Tukey’s Studentized Range test with a type I error of 0.05. Analyses were performed using SAS V9.4 for PC. 

## 5. Conclusions

Although our study showed good results in indoor food environments, new investigations, supported by a multidisciplinary approach, will make it possible to deepen the knowledge on this method of sanitation, also in light of any future epidemic/pandemic waves.

## Figures and Tables

**Table 1 pathogens-11-00608-t001:** Air samples in supermarket with ozone (A–B) and without ozone (C–D). Median (range) of bacterial and fungal count (cfu/m^3^) stratified by hours and type of supermarket. Marginal values are global median of main effect.

**Supermarket**	**Air Total Bacterial Count (cfu/m^3^)**		
	**8:00 am**	**12:00 pm**	**Total**
A–B	28 (0–83) ^c^	72.5 (44–170)	44.5 (0–170) ^a^
C–D	93 (39–264) ^c^	87 (44–183)	87.5 (39–264) ^a^
A–B–C–D	44 (0–264) ^b^	84.5 (44–183) ^b^	
**Supermarket**	**Air Total Fungal Count (cfu/m^3^)**		
	**8:00 am**	**12:00 pm**	**Total**
A–B	25 (0–440)	17 (0–50) ^d^	22 (0–440) ^e^
C–D	46.5 (11–180)	85.5 (28–344) ^d^	64.5 (11–344) ^e^
A–B–C–D	33 (0–440) ^f^	44.5 (0–344) ^f^	

^a^: *p* = 0.0042; ^b^*: p* = 0.0052; ^c^: *p* < 0.05; ^d^: *p* < 0.05; ^e^: *p* = 0.0002; ^f^: *p* = 0.0078.

**Table 2 pathogens-11-00608-t002:** Surface samples in supermarket with ozone (A–B) and without ozone (C–D). Median (range) of bacterial and fungal count (cfu/swab) by hours and type of supermarket. Marginal values are global median of main effect.

**Supermarket**	**Surface Total Bacterial Count (cfu/swab)**		
	**8:00 am**	**12:00 pm**	**Total**
A–B	0 (0–4)	0 (0-2)	0 (0–4) ^a^
C–D	8.5 (0–114)	9.5 (0-185)	9.5 (0–185) ^a^
A–B–C–D	0 (0–114)	1 (0-185)	
**Supermarket**	**Surface Total Fungal Count (cfu/swab)**		
	**8:00 am**	**12:00 pm**	**Total**
A–B	0.5 (0–12)	0 (0–5)	0 (0–12)
C–D	1.5 (0–14)	1 (0–5)	1 (0–14)
A–B–C–D	1 (0–14)	0.5 (0–5)	

^a^: *p* = 0.01.

## Data Availability

Not applicable.

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
