# Peer review of "Investigations on the Efficacy of Ozone as an Environmental Sanitizer in Large Supermarkets"

_pathogens, 2022, doi:10.3390/pathogens11050608_

Round 1

Reviewer 1 Report

General comment: The manuscript „ Investigations on the efficacy of ozone as environmental sanitizer in large supermarkets” by Caggiano and co-workers investigated the effect of Ozon in different supermarkets. As Ozon is already known as good antimicrobial, Caggiano and co-workers investigated the TBC and TFC in the air and on different surfaces after Ozon treatment in the supermarkets. The study concluded that statistical significant TBC and TFC reductions could be achieved with ozon treatment. The manuscript is suitable for publication in Pathogens, although some minor points need to be addressed.

Minor comments:

  • Line 22: Suggestion to remove "Background" 
  • Line 26: Suggestion to remove "Methods" 
  • Line 29: Suggestion to remove "Results" 
  • Line 32: Suggestion to remove "Conclusion" 

Author Response

We thank the Reviewer for comments that permitted us to  improve our paper

  • Line 22: Suggestion to remove "Background" 
  • Line 26: Suggestion to remove "Methods" 
  • Line 29: Suggestion to remove "Results" 
  • Line 32: Suggestion to remove "Conclusion" 

According to your suggestion, we removed “Background, Methods, Results and Conclusion

Reviewer 2 Report

Investigations on the efficacy of ozone as environmental sanitizer in large supermarkets

This is a straight forward investigation for the effectiveness of using ozone in large supermarket. Ozone has been used for an environmental sanitizer for a long time. However, no study was conducted in a commercial center, such as supermarket, due to the difficulty of gathering samples and conducting survey. This study conducted the bacterial and fungal populations in the air and surfaces. Results showed applying ozone was effective and significantly lower bacterial/fungal loads were obtained in the supermarkets with ozone application. However, some key elements are not presented, for example the concentrations of O3 in the supermarkets. In addition, the O3 concentrations in the supermarkets should be dynamic and varied through applying time and the sample collecting time. Furthermore, the discussion is too brief. Though this study did not involve complicated scientific techniques, it offered a significant tribute to epidemiology and public health. 

  1. Line 51: there are many physical methods, such as UV and pulse light, to inactivate microorganisms in a close environment. The advantage of physical methods is no residues. Authors could describe more techniques which are using presently and the advantages and disadvantages. Additionally, the major reason to use ozone.
  2. Line 87: suggest convert m3 to m3 throughout the manuscript.
  3. Line 89: Fungal load decreased from 8:00 am to 12:00 pm in C and D supermarkets. Do authors have any discussion?
  4. Line 103-109: Could authors provide some description for the prevalence of fungal species in the supermarkets? For example, why was Penicillium the predominate fungus?
  5. Line 104: Scientific names are not italic.
  6. Line 104-108: Are there are references related to the fungal species in environment? More comparison and discussion is needed.
  7. Line 117-124: TFC/swab between A/B and C/D was not significant. However, TFC and TBC in air were significantly different between A/B and C/D markets. Authors should provide some discussion for these results.
  8. Line 198: “2.64 g ozone per hour”, what was the concentration in the air?
  9. Line 203: “four consecutive days”, which four days? Saturday and Sunday or other holidays should have more customers than other days. Thus, which day is critical.
  10. Line 210: Only two sampling time. Did authors consider more sampling time, such as the closing hour? After a full day, the closing hour should have the highest microbial load. Did author conduct a preliminary study to confirm the change of microbial population during a working day?
  11. Line 148: agar solution? This confuse me. Additionally, 2% agar should be solid not liquid. Were the tiles on the top of 2% agar no tin the agar?
  12. Line 143: was 24-h incubation enough to form biofilm?
  13. Line 144: Authors describe the humidity is a key factor for ozone disinfection. However, did authors check the humidity in the supermarkets?
  14. Line 160: TBC was significantly different between the supermarkets but TFC was not. Could author provide more discussion for this.

Author Response

Reviewer 2

We thank the Reviewer for important comments that permitted us to  improve our paper.

This is a straight forward investigation for the effectiveness of using ozone in large supermarket. Ozone has been used for an environmental sanitizer for a long time. However, no study was conducted in a commercial center, such as supermarket, due to the difficulty of gathering samples and conducting survey. This study conducted the bacterial and fungal populations in the air and surfaces. Results showed applying ozone was effective and significantly lower bacterial/fungal loads were obtained in the supermarkets with ozone application. However, some key elements are not presented, for example the concentrations of O3 in the supermarkets. In addition, the O3 concentrations in the supermarkets should be dynamic and varied through applying time and the sample collecting time. Furthermore, the discussion is too brief. Though this study did not involve complicated scientific techniques, it offered a significant tribute to epidemiology and public health.

  1. Line 51: there are many physical methods, such as UV and pulse light, to inactivate microorganisms in a close environment. The advantage of physical methods is no residues. Authors could describe more techniques which are using presently and the advantages and disadvantages. Additionally, the major reason to use ozone.

            Thank you, we inserted at line 54-57

  1. Line 87: suggest convert m3 to m3 throughout the manuscript.

Thanks we convert m3 to m3 throughout the manuscript.

  1. Line 89: Fungal load decreased from 8:00 am to 12:00 pm in C and D supermarkets. Do authors have any discussion?

We inserted in the text at paragraph Discussion Lines 170-179

  1. Line 103-109: Could authors provide some description for the prevalence of fungal species in the supermarkets? For example, why was Penicillium the predominate fungus?

We inserted in the text at paragraph Discussion Lines 193-198

  1. Line 104: Scientific names are not italic.

Thank you, we change using Italic

  1. Line 104-108: Are there are references related to the fungal species in environment? More comparison and discussion is needed.

According to Reviewer we inserted more discussion. Paragraph Discussion lines 190-193.

  1. Line 117-124: TFC/swab between A/B and C/D was not significant. However, TFC and TBC in air were significantly different between A/B and C/D markets. Authors should provide some discussion for these results.

We inserted more discussion. Paragraph Discussion lines 199-205

  1. Line 198: “2.64 g ozone per hour”, what was the concentration in the air?

We thank the reviewer for this question which helps us to better clarify the activities of this study.

We added lines 233-238During the numerous samplings carried out at the four points of sale investigated, it was not possible to determine the concentration of airborne ozone in a punctiform manner by means of portable devices for measuring the O3 concentration. therefore this preliminary study offers a further point of investigation, as it would be interesting to evaluate the concentration during the whole day both in the air and on surfaces

  1. Line 203: “four consecutive days”, which four days? Saturday and Sunday or other holidays should have more customers than other days. Thus, which day is critical.

The consecutive days were midweek days, not holidays, in order to reproduce conditions as reproducible as possible and with medium turnout. We added this particular in the text  lines 260-262

  1. Line 210: Only two sampling time. Did authors consider more sampling time, such as the closing hour? After a full day, the closing hour should have the highest microbial load. Did author conduct a preliminary study to confirm the change of microbial population during a working day?

This represented a preliminary study to evaluate the change of microbial population during a working day. Other studies would security help us to confirm our results. However, we added the following sentences in the discussion to better precise this observation. (Lines 226-232)

  1. Line 148: agar solution? This confuse me. Additionally, 2% agar should be solid not liquid. Were the tiles on the top of 2% agar no tin the agar?

In our study we used only petri dishes, therefore containing solid agar medium and not in solution

  1. Line 143: was 24-h incubation enough to form biofilm?

In our study we did not evaluate the presence of biofilm but the total bacterial load, which also in accordance to the UNI EN ISO 4833-1:2013, is evaluated by incubating the petri plates for 72 hours.

  1. Line 144: Authors describe the humidity is a key factor for ozone disinfection. However, did authors check the humidity in the supermarkets?

Thank you for the opportunity to clarify this matter. We conducted the study in four supermarkets, substantially similar in terms of structure, lay-out, and plant characteristics related to the air conditioning and mechanical ventilation systems. In all supermarkets, both equipped with ozonation systems and the two not equipped, touch-screens were connected to the air conditioning and mechanical ventilation system control units. Using these devices, a room temperature between 20 and 22°C was set and detected, without being able to act on the quantity of relative humidity kept in the indoor environment.

In the Discussion paragraph, we added the following sentence at lines 159-164: “Through touch-screen displays connected to the air conditioning and mechanical ventilation system control units, microclimatic characteristics were set and detected inside the four points of sale. In particular, a temperature between 20 and 22°C was set and detected inside the four points of sale. However, these systems were not directly able to modify the quantity of relative humidity in the indoor environment.”

  1. Line 160: TBC was significantly different between the supermarkets but TFC was not. Could author provide more discussion for this.

We added in discussion paragraph (Line 197-198)

  1. The dynamic change of microbial load in supermarkets or other commercial establishments have been reported. Authors could cite these references.

Lines 180-184: The dynamic change of microbial load in supermarkets or other commercial establishments have been reported (Lee 2021, Boonrattanakij 2021), but to our knowledge, this is the first study that compare both bacterial and fungal contamination in supermarkets equipped with ozonation systems and super-markets without.

Round 2

Reviewer 2 Report

  1. It is better that authors provide the relative humidity and light intensity (lux) of the sampling sites of supermarkets since both are important for ozone degradation. 
  2. Since authors has provided a reference for the microflora in supermarkets, it is better that authors compare the fungal and bacterial species and load with the previous study.

Author Response

We thank the Referee for the comments that permitted us to  improve our paper.

 It is better that authors provide the relativfor e humidity and light intensity (lux) of the sampling sites of supermarkets since both are important for ozone degradation. 

we added this information in the Discussion paragraph, lines 214-217

Since authors has provided a reference for the microflora in supermarkets, it is better that authors compare the fungal and bacterial species and load with the previous study.

we added this information in the Discussion paragraph, lines 236-257